# Are there lost opportunities in chronic kidney disease? A region-wide cohort study

Johan Sundström,[1,2] Anna Norhammar,[3,4] Stelios Karayiannides,[5,6] Johan Bodegård,[7] Stefan Gustafsson,[8] Thomas Cars,[8] Maria Eriksson Svensson,[9,10] Johan Ärnlöv[11,12]

For numbered affiliations see end of article.

**Correspondence to**
Professor Johan Sundström;
johan.sundstrom@uu.se

## ABSTRACT

**Objectives** Identify the windows of opportunity for the diagnosis of chronic kidney disease (CKD) and the prevention of its adverse outcomes and quantify the potential population gains of such prevention.

**Design and setting** Observational, population-wide study of residents in the Stockholm and Skåne regions of Sweden between 1 January 2015 and 31 December 2020.

**Participants** All patients who did not yet have a diagnosis of CKD in healthcare but had CKD according to laboratory measurements of CKD biomarkers available in electronic health records.

**Outcome measures** We assessed the proportions of the patient population that received a subsequent diagnosis of CKD in healthcare, that used guideline-directed pharmacological therapy (statins, renin-angiotensin aldosterone system inhibitors (RAASi) and/or sodium-glucose cotransporter-2 inhibitors (SGLT2i)) and that experienced adverse outcomes (all-cause mortality, cardiovascular mortality or major adverse cardiovascular events (MACE)). The potential to prevent adverse outcomes in CKD was assessed using simulations of guideline-directed pharmacological therapy in untreated subsets of the study population.

**Results** We identified 99 382 patients with undiagnosed CKD during the study period. Only 33% of those received a subsequent diagnosis of CKD in healthcare after 5 years. The proportion that used statins or RAASi was of similar size to the proportion that didn't, regardless of how advanced their CKD was. The use of SGLT2i was negligible. In simulations of optimal treatment, 22% of the 21 870 deaths, 27% of the 14 310 cardiovascular deaths and 39% of the 22 224 MACE could have been avoided if every patient who did not use an indicated medication for their laboratory-confirmed CKD was treated with guideline-directed pharmacological therapy for CKD.

**Conclusions** While we noted underdiagnosis and undertreatment of CKD in this large contemporary population, we also identified a substantial realisable potential to improve CKD outcomes and reduce its burden by treating patients early with guideline-directed pharmacological therapy.

## STRENGTHS AND LIMITATIONS OF THIS STUDY

⇒ This study was strengthened by its access to health registries and electronic health records detailing most incidences of healthcare and medication use by residents in two of Sweden's largest regions, representing over a third of the national population.

⇒ Given that only persons with suspected or known other diseases have provided the samples necessary to make a laboratory-based assessment of chronic kidney disease (CKD) status, it is likely that this study underestimates the number of patients with undiagnosed CKD.

⇒ This study could not assess if patients from a specific socioeconomic background or ethnicity were disproportionally affected by the deficiencies in CKD-related healthcare.

⇒ Simulations of guideline-directed pharmacological therapy assumed that treatment was initiated at the time CKD was identified using laboratory measurements and that the adherence to treatment would be similar to that obtained in clinical trials, efficiency and success in treatment that are unlikely in routine clinical practice.

700 million and 1 billion people globally).[1–3] As the prevalence of CKD increases with an ageing population and growing rates of CKD risk factors (eg, diabetes and hypertension),[4] so too does its contribution to total mortality and its economic burden[2 5 6]; highlighting the need to evaluate CKD diagnosis and treatment patterns.

Given that cardiovascular disease is a major cause of premature death among patients with CKD, management of cardiovascular disease risk factors (eg, high cholesterol, hypertension and chronic hyperglycaemia) remains central to the treatment of CKD. Statins, renin-angiotensin aldosterone system inhibitors (RAASi) and sodium-glucose cotransporter-2 inhibitors (SGLT2i) are first-line pharmacological therapies in patients with CKD[7–9]; each has been demonstrated to reduce the risk of major adverse

## INTRODUCTION

Current estimates indicate that 1 in 10 adults has chronic kidney disease (CKD; between

cardiovascular events (MACE) in this patient population (online supplemental table S1). Data from recent large cohort studies suggest that low proportions of patients with any degree of laboratory-confirmed CKD are diagnosed with CKD or treated with these drugs.[1 10 11] While low use of SGLT2i may be explained by the recency of their inclusion in CKD treatment guidelines,[11–13] it is less clear as to why statins and RAASi have been underused, considering that they have been readily available in clinical practice for more than 20 years. Hence, detailed characterisation of the prevalence and consequences of suboptimal CKD-related prevention and healthcare and determination of the potential for improving outcomes in patients with CKD are of great public health importance.

For that purpose, we identified a large cohort of patients with CKD using laboratory data in electronic health records, with the aim of detailing the proportions of those patients who received a subsequent diagnosis of CKD in healthcare, who used guideline-directed pharmacological therapy for their CKD and who developed adverse outcomes. We also aimed to identify windows of opportunity for preventing adverse outcomes in patients with CKD by assessing the absolute number of adverse outcomes that would be prevented by the simulated application of several variants of guideline-directed pharmacological therapy.

## METHODS
### Data sources

Sweden has a comprehensive, nationwide public healthcare system to which each resident has access with a minor co-payment for healthcare visits, hospitalisations and medications.[14] Residents have a unique personal identification number (person-ID),[15] which is mandatory for all administrative purposes, including any contact with the healthcare system and filling of drug prescriptions. Thus providing a basis for a complete population-wide medical history.

The present study used data from the CELOSIA database, the details of which can be viewed in online supplemental file 1. Briefly, it includes data from the Swedish Prescribed Drug Register, the Cause of Death Register, the National Patient Register and several data sources with regional coverage, including electronic health records in the Stockholm and Skåne regions. A complete list of the Anatomical Therapeutic Chemical (ATC) codes,[16] International Classification of Diseases (ICD)-10 codes,[17] procedure codes[18] and laboratory values used in the CELOSIA database is available in online supplemental table S2.

Individual patient-level data from the national registers and regional data sources were linked using the person-ID. The linked pseudonymised database was managed separately by Sence Research AB, Uppsala, Sweden. The study was approved by the Swedish Ethical Review Authority (approval numbers 2020-03850 and 2020-06716).

### Study population

Participant flow through the study is described in online supplemental figure S1. Laboratory measurements of creatinine-based estimated glomerular filtration rate (eGFR) and/or urine albumin-creatinine ratio (UACR), available in the electronic health records, were used to identify patients >18 years of age with CKD (irrespective of whether it was diagnosed, indicated by the presence of an ICD code in healthcare databases) according to the Kidney Disease: Improving Global Outcomes (KDIGO) clinical practice guidelines.[8] Meaning, CKD was identified if a patient presented at least two abnormal eGFR measurements (<60 mL/min/1.73 m$^2$) and/or two abnormal UACR measurements (≥3 g/mol), with the first and last measurements taken at least 90 days apart and where all measurements in between were abnormal.

The last abnormal measurement, which would have allowed the identification of CKD, needed to have been taken between 1 January 2015 and 31 December 2020. The first abnormal measurement, however, could have been taken prior to the start of that study period. The date on which CKD was identified was considered the index date. The revised Lund-Malmö equation was used to calculate eGFR from creatinine, age and sex,[19] and UACR was expressed in g/mol. The eGFR and UACR thresholds defining each stage of CKD according to KDIGO guidelines are detailed in online supplemental table S3.

At the time of and within the 18 months prior to the date that CKD was identified (index), patients needed to be residing in Region Stockholm or Region Skåne where electronic health records were available. Patients with a prior diagnosis of CKD in healthcare or who had donated a kidney were excluded (ICD-10 codes used to identify those patients are listed in online supplemental table S4; ICD-10 codes for a broad CKD definition were used). Patients with zero days of follow-up (ie, those who died at the time CKD was identified) were also excluded (n=92).

### Follow-up and characterisation of exposures, outcomes and comorbidities

Patients were followed for up to 5 years or until the censor date (31 December 2020), whichever occurred first. Adverse outcomes of interest during follow-up were all-cause death, non-cardiovascular death and MACE, where MACE was defined as a primary diagnosis of MACE in inpatient care or a primary/non-primary diagnosis of death caused by cardiovascular disease.

To evaluate the lapse in time between the laboratory-based identification of CKD and a subsequent diagnosis of CKD in healthcare, ICD-10 codes indicating a CKD diagnosis in healthcare were also searched for. Prevalent comorbidities were defined as all diagnoses up until the laboratory-based identification of CKD. Diagnoses available from primary care, specialised outpatient care and inpatient care were used, including both primary and non-primary diagnoses. The ICD-10 codes used to identify diagnoses of CKD in healthcare (codes for a narrow CKD definition were used), prevalent comorbidities and

**Table 1** Baseline characteristics of the total study cohort with laboratory-identified CKD and that stratified by KDIGO-defined stages of CKD

| | Total | S3a eGFR 45–59 mL/min/1.73 m² | S3b eGFR 30–44 mL/min/1.73 m² | S4+S5 eGFR<30 mL/min/1.73 m² | A2 UACR 3–30 g/mol | A3 UACR >30 g/mol |
|---|---|---|---|---|---|---|
| N | 99 382 | 72 407 | 14 315 | 1740 | 9549 | 1371 |
| Female, N (%) | 51 198 (51.5%) | 37 477 (51.8%) | 8465 (59.1%) | 1180 (67.8%) | 3532 (37.0%) | 544 (39.7%) |
| Age, median (IQR) | 77.7 (71.3–84.2) | 77.9 (72.3–83.6) | 83.4 (76.7–88.8) | 87.3 (81.5–91.6) | 65.3 (56.0–73.2) | 62.0 (50.5–70.4) |
| Socioeconomic status, N (%) | | | | | | |
| Missing | 65 101 (65.5%) | 46 897 (64.8%) | 10 187 (71.2%) | 1342 (77.1%) | 5757 (60.3%) | 918 (67.0%) |
| High | 7354 (21.5%) | 5847 (22.9%) | 881 (21.3%) | 85 (21.4%) | 497 (13.1%) | 44 (9.7%) |
| Middle | 14 154 (41.3%) | 10 752 (42.1%) | 1656 (40.1%) | 135 (33.9%) | 1444 (38.1%) | 167 (36.9%) |
| Low | 12 773 (37.3%) | 8911 (34.9%) | 1591 (38.5%) | 178 (44.7%) | 1851 (48.8%) | 242 (53.4%) |
| Comorbidities, N (%) | | | | | | |
| Heart failure | 20 290 (20.4%) | 14 511 (20.0%) | 4274 (29.9%) | 625 (35.9%) | 744 (7.8%) | 136 (9.9%) |
| Coronary heart disease | 26 611 (26.8%) | 19 555 (27.0%) | 4543 (31.7%) | 555 (31.9%) | 1701 (17.8%) | 257 (18.7%) |
| Acute myocardial infarction | 11 285 (11.4%) | 8197 (11.3%) | 1999 (14.0%) | 255 (14.7%) | 725 (7.6%) | 109 (8.0%) |
| Unstable angina | 5498 (5.5%) | 4075 (5.6%) | 829 (5.8%) | 114 (6.6%) | 424 (4.4%) | 56 (4.1%) |
| Stroke | 16 717 (16.8%) | 12 256 (16.9%) | 2982 (20.8%) | 370 (21.3%) | 974 (10.2%) | 135 (9.8%) |
| Atrial fibrillation | 25 374 (25.5%) | 19 035 (26.3%) | 4515 (31.5%) | 575 (33.0%) | 1099 (11.5%) | 150 (10.9%) |
| Peripheral artery disease | 9153 (9.2%) | 6649 (9.2%) | 1543 (10.8%) | 184 (10.6%) | 676 (7.1%) | 101 (7.4%) |
| Diabetes | 31 745 (31.9%) | 19 510 (26.9%) | 4019 (28.1%) | 441 (25.3%) | 6866 (71.9%) | 909 (66.3%) |
| Cancer | 30 817 (31.0%) | 23 735 (32.8%) | 4652 (32.5%) | 571 (32.8%) | 1664 (17.4%) | 195 (14.2%) |
| Measurements, median (IQR) | | | | | | |
| Systolic blood pressure (mmHg) | 140.0 (127.0–150.0) | 140.0 (126.0–150.0) | 140.0 (125.0–153.0) | 140.0 (125.0–155.0) | 140.0 (130.0–150.0) | 140.0 (130.0–153.0) |
| Body mass index (kg/m²) | 26.5 (23.5–30.1) | 26.2 (23.4–29.7) | 25.8 (22.8–29.5) | 25.6 (22.6–29.3) | 29.0 (25.4–33.1) | 29.4 (25.4–33.7) |
| Laboratory, median (IQR) | | | | | | |
| P-creatinine (µmol/L) | 94.0 (81.0–108.0) | 93.0 (82.0–105.0) | 116.0 (102.0–133.0) | 151.0 (133.0–178.0) | 69.0 (59.0–80.0) | 70.0 (58.2–83.0) |
| eGFR (mL/min/1.73 m²) | 53.7 (46.9–57.9) | 54.1 (49.8–57.3) | 38.9 (33.8–43.0) | 24.5 (20.9–28.3) | 78.9 (70.2–90.1) | 80.0 (70.6–92.8) |
| UACR (g/mol) | 4.0 (1.0–9.7) | 1.3 (0.5–4.6) | 2.5 (0.7–9.6) | 6.6 (2.1–27.6) | 6.6 (4.3–12.0) | 54.1 (34.0–90.0) |
| Sodium (mmol/L) | 140.0 (138.0–142.0) | 140.0 (138.0–142.0) | 140.0 (138.0–142.0) | 140.0 (138.0–142.0) | 140.0 (138.0–141.0) | 139.0 (137.0–141.0) |
| Potassium (mmol/L) | 4.2 (3.9–4.4) | 4.2 (3.9–4.4) | 4.2 (4.0–4.5) | 4.3 (4.0–4.7) | 4.1 (3.9–4.3) | 4.1 (3.9–4.3) |
| Glycated haemoglobin A1c (mmol/mol) | 44.0 (39.0–55.0) | 42.0 (38.0–55.0) | 44.0 (38.0–55.0) | 44.0 (38.0–57.0) | 53.0 (44.0–67.0) | 53.0 (42.0–69.0) |
| Haemoglobin (g/L) | 134.0 (123.0–145.0) | 135.0 (124.0–145.0) | 128.0 (117.0–139.0) | 122.0 (111.0–132.0) | 142.0 (131.0–152.0) | 140.0 (128.0–153.0) |

Continued

**Table 1** Continued

| | Total | S3a eGFR 45–59 mL/min/1.73 m² | S3b eGFR 30–44 mL/min/1.73 m² | S4+S5 eGFR<30 mL/min/1.73 m² | A2 UACR 3–30 g/mol | A3 UACR >30 g/mol |
|---|---|---|---|---|---|---|
| Medications, N (%) | | | | | | |
| Statins | 40 732 (41.0%) | 29 271 (40.4%) | 5594 (39.1%) | 625 (35.9%) | 4611 (48.3%) | 631 (46.0%) |
| Renin-angiotensin aldosterone system inhibitors | 56 770 (57.1%) | 40 626 (56.1%) | 8460 (59.1%) | 972 (55.9%) | 5854 (61.3%) | 858 (62.6%) |
| Sodium-glucose cotransporter-2 inhibitors | 986 (1.0%) | 470 (0.6%) | 52 (0.4%) | ≤5 | 412 (4.3%) | 49 (3.6%) |

Data for baseline demographics and prevalent comorbidities were collected for each patient on the date this study identified their CKD (index date) using laboratory measurements of eGFR and UACR according to the KDIGO clinical practice guidelines. The most recent clinical measurements and laboratory values within the 1 year prior to the index date are reported. For patients residing in Region Stockholm, the most recent socioeconomic data within the 10 years prior to the index date was collected from the Mosaic system, which applies the principles of geodemography to consumer household and individual data to categorise patients into one of the following three levels: (1) highest income/education; (2) medium income/education and (3) lowest income/education. Stratification of the patients into the different KDIGO-defined stages of CKD using eGFR (stages S3a, S3b and S4+S5) and UACR measurements (stages A2 and A3) is based on the median of all measurements in the ≥90-day period required to diagnose CKD.

CKD, chronic kidney disease; eGFR, estimated glomerular filtration rate; KDIGO, Kidney Disease: Improving Global Outcomes; UACR, urine albumin–creatinine ratio.

adverse outcomes of interest are listed in online supplemental table S4.

To characterise the use of guideline-directed pharmacological therapy within the total study population, filled prescriptions for statins, RAASi and SGLT2i were searched for using the ATC codes listed in online supplemental table S5. For each medication, a patient was considered treated throughout the follow-up period if a new prescription was filled before the coverage period of the previously filled prescription expired. The coverage period for each filled prescription was calculated as the total number of pills dispensed divided by the prescribed daily dosage. This coverage period was extended by a 25% grace period, allowing for small deviations (eg, potentially due to imperfect medication adherence) from the expected dispensation pattern.

### Statistical analyses

All statistical analyses were performed using R V.3.6.0, and the R package Epi was used for all multistate analyses.[20 21]

### Proportions of patients with diagnoses, treatments and adverse outcomes

To detail the proportion of patients with laboratory-identified CKD that received a subsequent diagnosis of CKD in healthcare and/or used guideline-directed pharmacological therapy (statins, RAASi and/or SGLT2i), as well as the proportion affected by adverse outcomes, multistate models were constructed with four possible transient states:

1. Undiagnosed and untreated.
2. Undiagnosed and treated.
3. Diagnosed and untreated.
4. Diagnosed and treated.

The transient nature of the multistate models meant that any change in a patient's state during follow-up was accounted for by the model and reflected in the results (eg, any change in treatment status over time or at the first diagnosis of CKD in healthcare). Each model also included either all-cause death, non-cardiovascular disease death or MACE as terminal, absorbing states. The Aalen-Johansen estimator of transition probabilities was used in a multistate model adjusting for age (categorical (18–45) (45–65) (65–80) (80–inf)) and sex.

In figures for the multistate models and all tables where patients are stratified by the severity of their CKD, patients were stratified primarily using eGFR measurements. If eGFR data were normal or unavailable, UACR measurements were used to stratify the patients. To detail the lapse in time between the laboratory-based identification of CKD and a diagnosis of CKD in healthcare, treatment and/or an adverse event, the Aalen-Johansen estimator was calculated along the time scale, years since the laboratory-based identification of CKD. Kaplan-Meier curves were used to present the proportions of patients who received a diagnosis of CKD in healthcare after the laboratory-based identification of CKD.

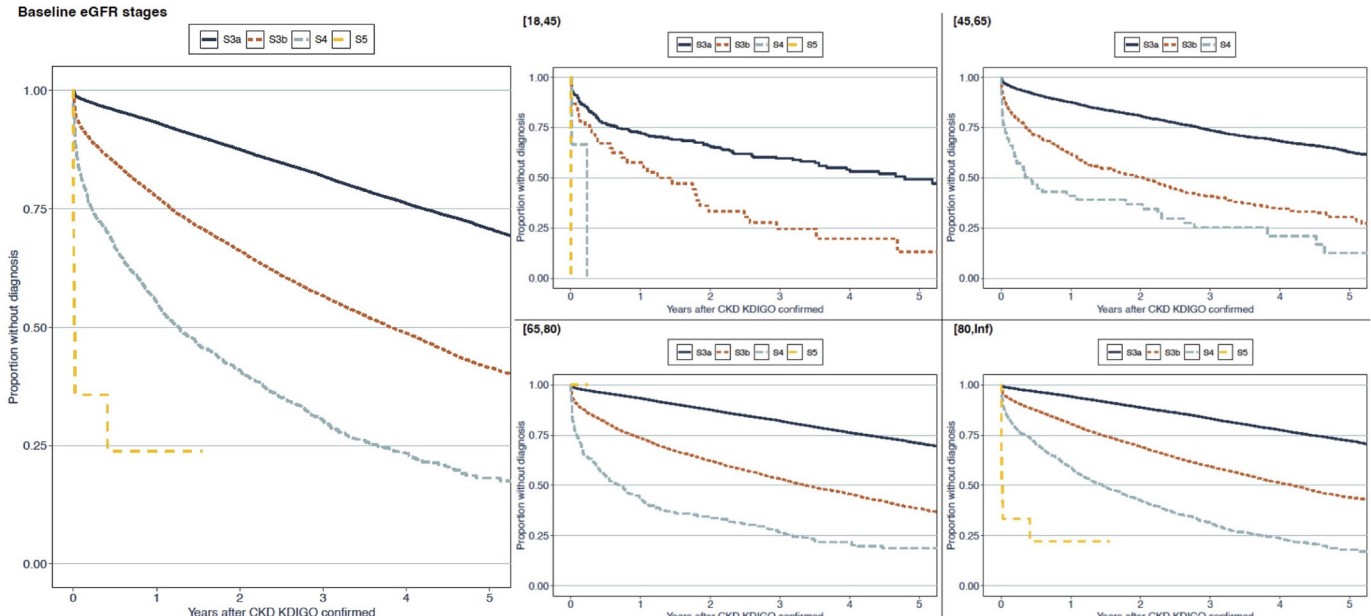

**Figure 1** Time to the first CKD diagnosis in healthcare after CKD was identified using measurements of eGFR. Kaplan-Meier curves. Patients are stratified by the Kidney Disease: Improving Global Outcomes-defined stage of their CKD and by age group. S3a indicates eGFR 45–59 mL/min/1.73 m$^2$; S3b indicates eGFR 30–44 mL/min/1.73 m$^2$; S4 indicates eGFR 15–29 mL/min/1.73 m$^2$ and S5 indicates eGFR <15 mL/min/1.73 m$^2$. CKD, chronic kidney disease; eGFR, estimated glomerular filtration rate.

## Simulations of the population effects of guideline-directed pharmacological therapy

To evaluate the potential to prevent adverse outcomes in patients with CKD, several variants of guideline-directed pharmacological therapy were simulated in non-overlapping subsets of the study population, which included all patients who were not treated with one or more of the statins, RAASi or SGLT2i, throughout follow-up. For example, in simulations of statin therapy, patients included in the analysis could not have been treated with statins during the entire follow-up period, but they could have been treated with RAASi or SGLT2i. Separate analyses were conducted for monotherapy with each treatment and for all possible combinations of treatments in polytherapy (eg, any two of the medications of interest or all three medications in combination).

HRs from several large systematic reviews and meta-analyses,[22–24] which provided a comprehensive compilation of randomised controlled trials that evaluated the effect of statin, RAASi or SGLT2i treatment on all-cause death, cardiovascular death and MACE in patients with CKD, were used to calculate the probability of 5-year survival for those outcomes, simulating the use of guideline-directed pharmacological therapy in the total number of patients untreated with that specific therapy. From the difference between the actual probability of an event in the untreated population, as observed during follow-up, and the estimated probability of 5-year survival, $S_{untreated}(5y)$ and $S_{treated}(5y)$, the absolute reduction in risk of an adverse event, $\Delta_{risk}$, was calculated. Using this value, the number needed to treat, *NNT*, to prevent one adverse

event, the number of prevented deaths (by any cause) and MACE were estimated for each treatment.

$$S_{treated}(5y) = S_{untreated}(5y)^{\exp(log_n(HR_{RAASi})+log_n(HR_{statin})+log_n(HR_{SGLT2i}))};$$

$$\Delta_{risk} = (1 - S_{untreated}(5y)) - (1 - S_{treated}(5y));$$

$$NNT = 1/\Delta_{risk};$$

$$N_{saved} = N_{untreated}/NNT$$

It was assumed that the treatment effect reported by each meta-analysis was replicable in each simulation and that medication adherence was similar to that obtained in clinical trials from the date that CKD was identified using laboratory measurements. For simulations of monotherapy, the relevant HR extracted from each meta-analysis was used. For combination therapy, HRs were calculated using the HRs extracted from the three meta-analyses under the assumption that there was no deviation from a multiplicative relative effect of using treatments in combination (ie, no interaction between medications), as shown previously.[25–27] Multiple sensitivity analyses were conducted to account for variable treatment effects due to differences in the population, including a subgroup of younger age (with a higher likelihood to be considered for preventive treatment), a subgroup with type-2 diabetes or hypertension (which would already have other indications for some of the drugs and hence be more likely to be considered for preventive treatment) a subgroup that survived more than 90 days after index (which would be more likely to be considered for preventive treatment than very sick patients), and variable treatment effects due to differences in

the treatment, including combinations of assumptions of multiplicative effects of all drugs assumed, but HR for SGLT2i set to 1, and assumptions of no multiplicative effects, only the HR from the drug with the largest effect counted, HR for all other drugs set to 1. These analyses also investigated the influence of less-than-optimal treatment effect (the intention-to-treat effect obtained in clinical trials); that is, if there are delays in initiating treatment, adherence is less than that obtained in clinical trials and/or other reasons for not expecting optimal treatment effect, a number less than 100 could be considered.

### Patient and public involvement

Patients and the public were not involved in the design, conduct, reporting, or dissemination plans of this study.

## RESULTS
### Baseline characteristics

In a background population of approximately 3.8 million residents between the beginning of 2015 and the end of 2020, a total of 99 382 patients without a medical record of CKD were identified as having CKD using laboratory measurements of eGFR and/or UACR. The characteristics of these patients are presented in table 1 and online supplemental table S6. Overall, patients were elderly (median age: 77.7 years; IQR: 71.3–84.2 years), and the prevalence of cardiovascular diseases and diabetes was high. At baseline, those with CKD stages A2 and A3 were younger, had a higher body mass index, were more often male,

had diabetes and had more preventive drug treatment than those with stages S3a–S5 (table 1). Patients were followed for a median of 3 years (IQR: 1.6–4.5 years).

### Two out of three patients with CKD remain undiagnosed after 5 years

After 5 years from the date that CKD was identified using laboratory measurements, only 33% of the 99 382 patients received a subsequent diagnosis of CKD in healthcare. Of that 33%, the median time to a diagnosis was 1.4 years (IQR: 0.4–2.7 years). The proportion of patients that received a subsequent diagnosis of CKD in healthcare varied between stages of KDIGO-defined CKD, largely irrespective of age (figures 1 and 2). Notably, the rates over time of receiving a CKD diagnosis were similar in stages S3a and A2 and stages S3b and A3, respectively (figures 1 and 2).

In patients with less advanced CKD (KDIGO-defined CKD stage 3a), identified using eGFR measurements, the proportion of patients that received a diagnosis of CKD in healthcare was lower than in those with more advanced CKD (KDIGO stages 4 and 5 CKD) (29% vs 82%). A similar pattern was found in those with CKD identified using UACR measurements (KDIGO stage A2 CKD, 34% vs KDIGO stage A3 CKD, 62%; online supplemental table S7). Analogously, the time to a diagnosis of CKD in healthcare was shorter in patients with more advanced stages of CKD.

### Use of guideline-directed pharmacological therapy in relation to a CKD diagnosis

The proportions of patients, stratified by the severity of their CKD according to eGFR or UACR measurements, who received a CKD diagnosis in healthcare or who

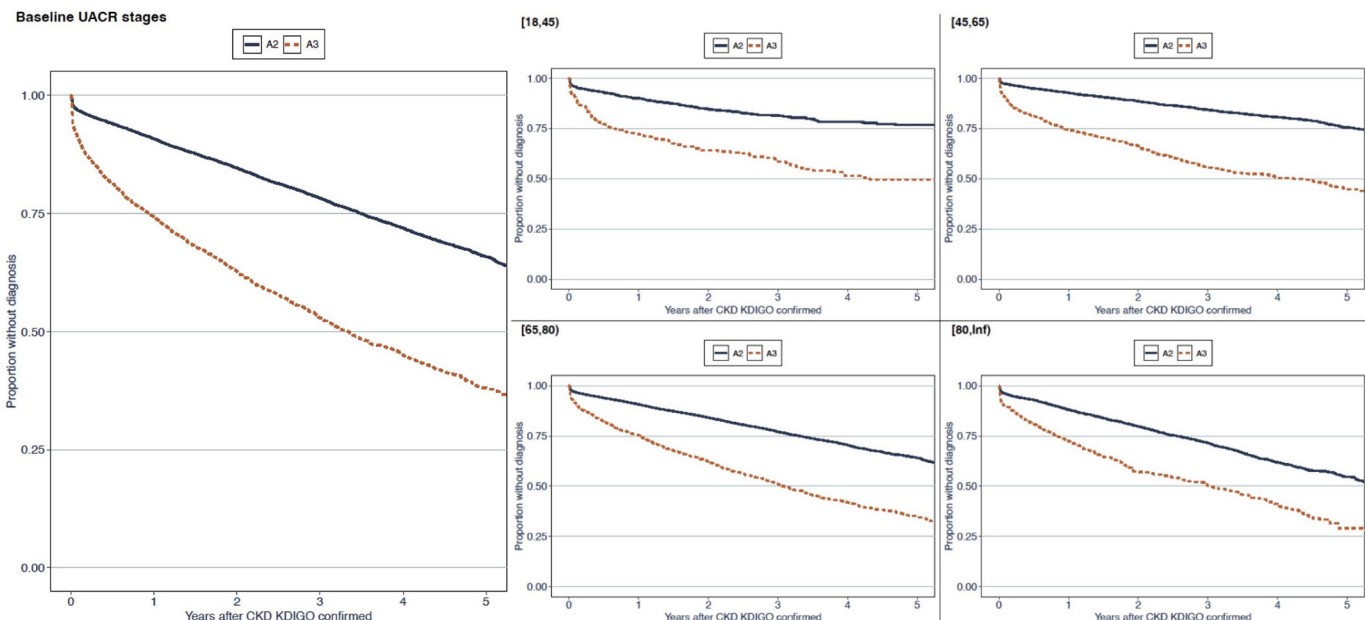

**Figure 2** Time to first CKD diagnosis in healthcare after CKD was identified using measurements of UACR. Kaplan-Meier curves. Patients are stratified by the KDIGO-defined stage of their CKD and by age group. All patients from the total cohort whose CKD could be identified using abnormal UACR measurements according to the KDIGO guidelines were included in these analyses, regardless of whether measurements of estimated glomerular filtration rate were available. A2 indicates UACR 3–30 g/mol and A3 indicates UACR >30 g/mol. CKD, chronic kidney disease; KDIGO, Kidney Disease: Improving Global Outcomes; UACR, urinary albumin-creatinine ratio.

used guideline-directed pharmacological therapy are presented in figure 3. Interestingly, around half of the patients were already using statins or RAASi when CKD was identified using laboratory measurements. The use of SGLT2i was negligible, as anticipated.

## Use of combination guideline-directed pharmacological therapy might greatly reduce absolute adverse outcome rates

After 5 years, 21 870 people died (of any cause), 14 310 people died from cardiovascular disease and 22 442 persons experienced a MACE. The proportion of patients that died from any cause or had a MACE increased with each worsening stage of CKD (online supplemental figures S2 and S3). Similarly, the proportions of those adverse outcomes increased with each incremental increase in age (online supplemental figures S4 and S5). Findings were similar between men and women.

In simulated monotherapy of statins, RAASi or SGLT2i reduced the incidences of all-cause death, cardiovascular death and MACE by 6%–23% (table 2). When variations of dual therapy were simulated, the incidences of all-cause death, cardiovascular death and MACE were reduced by 14%–35%. The greatest reductions in the incidences of all-cause death, cardiovascular death and MACE were estimated when all three pharmacological treatments were used in combination, preventing 25% of all-cause deaths, 26% of cardiovascular deaths and 46% of MACE. If every patient who did not use an indicated medication for their laboratory-confirmed CKD was treated according to CKD treatment guidelines, 5014 (22%) of the 21 870 all-cause deaths, 3871 (27%) of the 14 310 cardiovascular deaths and 8722 (39%) of the 22 442 MACE could have potentially been avoided. If everyone who was not treated with triple therapy had received it, the number needed to be treated to avoid one all-cause death, one cardiovascular death and one MACE would be 11, 15 and 7, respectively.

The sensitivity analysis showed that the potential to prevent all-cause deaths, cardiovascular deaths and MACE decreased as the proportion of patients who received optimal treatment decreased (figure 4). The potential to prevent these events was most notably lower in younger patients and when treatment with SGLT2i was not included as part of combination therapy.

## DISCUSSION

In a background population of approximately 3.8 million people from Sweden's Stockholm and Skåne regions, this study identified 99 382 patients with undiagnosed CKD in a 6-year period that ended in 2020. Of that population, CKD remained undiagnosed by healthcare in two out of three patients 5 years after CKD could be identified using available laboratory measurements. Those with more advanced CKD were diagnosed faster than those with less severe CKD, seemingly irrespective of age. The use of guideline-directed pharmacological therapy was not dramatically affected by a given CKD diagnosis, seemingly irrespective of the severity of the CKD, and there was very

little use of SGLT2i across the entire study population, as anticipated during this observation period. Simulations of guideline-directed pharmacological therapy indicated that there is potential to substantially improve outcomes in CKD, with the greatest projected protective effects resulting from the use of statins, RAASi and SGLT2i in combination.

### What opportunities are currently lost in CKD?

A CKD diagnosis would be anticipated to lead to several changes in care, such as more structured follow-up, changed dosing of several drug classes and more preventive measures taken in contrast to radiological examinations, to name but a few.

No striking changes to treatment regimens were observed around the time of a diagnosis of CKD in healthcare, suggesting that clinical practice did not always follow the clinical practice guidelines informing practitioners during the study period, such as those published by KDIGO in 2012 and 2013,[8 9 11] and national guidelines and recommendations. Indeed, the use of such pharmacological treatments is pertinent to this study's population, where the proportion of patients that were affected by MACE was far greater than the proportion that died of non-cardiovascular causes. The underuse of guideline-directed pharmacological therapy is consistent with recent large cohort studies, which also reported that only small proportions of patients with CKD were using statins, RAASi or SGLT2i.[1 10 11] Given that SGLT2i has only recently been included in clinical practice guidelines after the study period,[7 11–13] the low use of these agents was in line with expectations. Patients with CKD stages A2 and A3 more often used SGLT2i than those with stages S3a to S5, both at baseline and during follow-up (table 1 and figure 3), explained by their more than twice as high prevalence of diabetes (table 1). In sum, our data clearly show that opportunities are lost in both the diagnosis and treatment of CKD.

It should be noted that diagnosing and treating CKD is likely less incentivised in Sweden than in many other countries. A CKD diagnosis has smaller effects on rewards for healthcare units in Sweden, where a government-funded healthcare system covers all citizens than it would have under insurance or claims systems.[28] Furthermore, some other countries with similar healthcare systems have much stronger encouragement for registering a CKD diagnosis and providing evidence-based CKD care. The UK, for example, has had a focus on registering CKD diagnoses and providing evidence-based preventive care since 2006 as part of the Primary Care Quality and Outcomes Framework.[29]

### What are the windows of opportunity in CKD?

Simulations of pharmacological treatments indicated that, with greater adherence to guidelines by the caring physician and adequate medication adherence by the patient, there is significant potential to improve outcomes in CKD. Addressing cardiovascular disease risk is central to

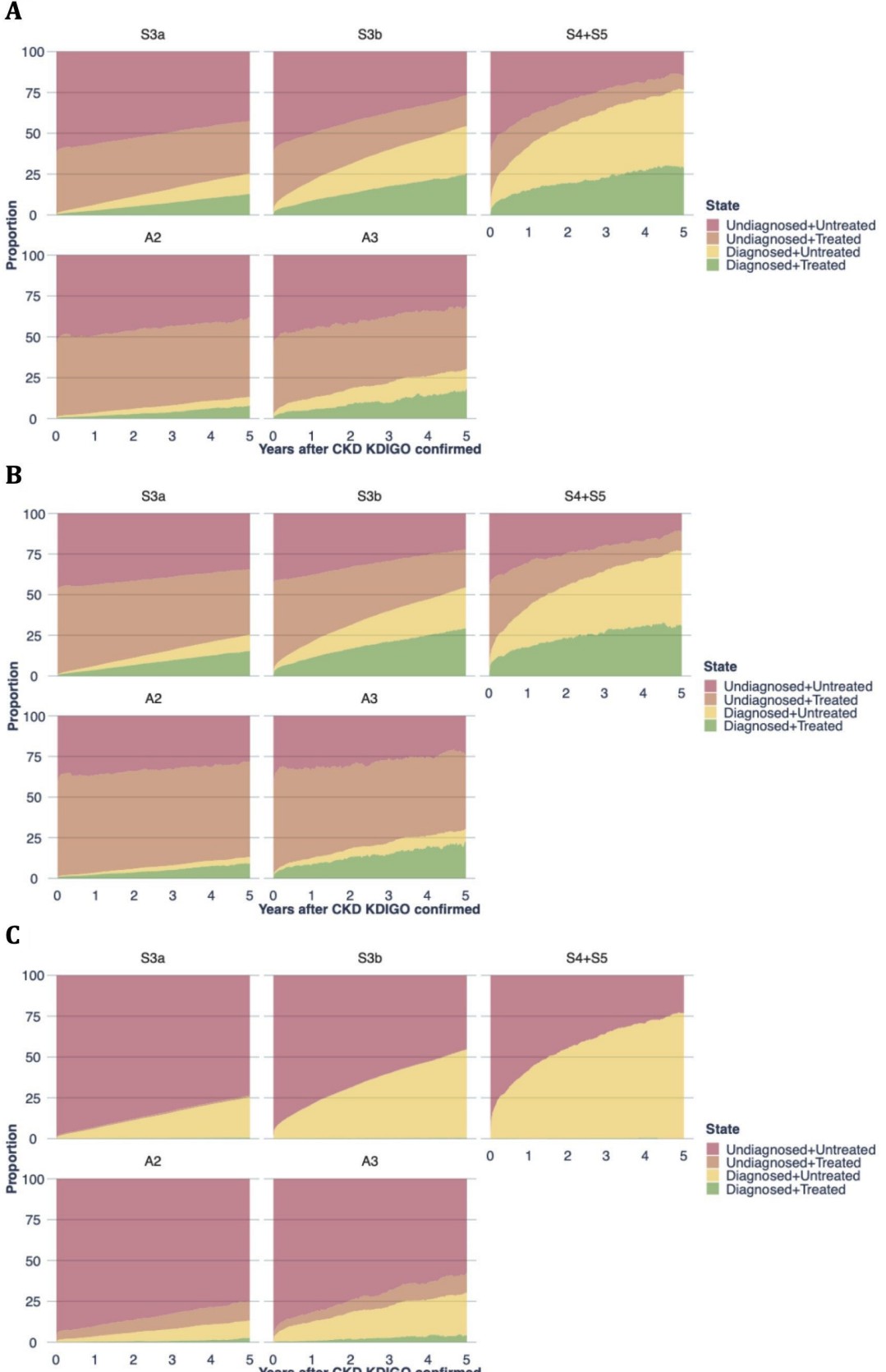

**Figure 3** Multistate models presenting the proportions of patients diagnosed or not diagnosed with CKD in healthcare that used or did not use guideline-directed pharmacological therapy. The use of (A) statins, (B) renin-angiotensin aldosterone system inhibitors and (C) sodium-glucose cotransporter-2 inhibitors was assessed. Patients are stratified by the KDIGO-defined stage of their CKD, identified using estimated glomerular filtration rate (stages S3a, S3b and S4+S5) and urinary albumin-creatinine ratio (stages A2 and A3). CKD, chronic kidney disease; KDIGO, Kidney Disease: Improving Global Outcomes.

**Table 2** Number of adverse events potentially prevented over 5 years in simulated provision of monotherapy and combination pharmacological therapy, in optimal settings, to all that were untreated with those corresponding regimens

| Treatment | Number of patients without specific treatment regimen, additional possibility to treat | Outcome (number of events) | Probability of an event in untreated | HR from meta-analyses | Estimated probability of an event | Absolute difference in risk | Number needed to treat | Events prevented |
|---|---|---|---|---|---|---|---|---|
| Eligible for additional monotherapy with: | | | | | | | | |
| Statins | 402 | Death[23] | 0.084 | 0.92 (0.85–0.99) | 0.077 | 0.007 | 156 | 3 |
| RAASi | 353 | Death[11] | 0.074 | 0.94 (0.84–1.07) | 0.070 | 0.004 | 234 | 2 |
| SGLT2i | 38 760 | Death (6415) | 0.250 | 0.87 (0.80–0.95) | 0.221 | 0.029 | 35 | 1106 |
| Statins | 402 | CVD death[18] | 0.065 | 0.91 (0.84–0.99) | 0.060 | 0.005 | 175 | 2 |
| RAASi | 353 | CVD death[4] | 0.018 | 0.94 (0.78–1.04) | 0.017 | 0.001 | 935 | 0 |
| SGLT2i | 38 760 | CVD death (4603) | 0.188 | 0.87 (0.79–0.95) | 0.165 | 0.023 | 45 | 862 |
| Statins | 400 | MACE (47) | 0.199 | 0.77 (0.70–0.85) | 0.157 | 0.042 | 24 | 17 |
| RAASi | 345 | MACE[14] | 0.083 | 0.84 (0.78–0.91) | 0.070 | 0.013 | 78 | 4 |
| SGLT2i | 36 766 | MACE (7477) | 0.294 | 0.84 (0.75–0.94) | 0.254 | 0.040 | 25 | 1488 |
| Eligible for additional dual therapy with: | | | | | | | | |
| Statins+RAASi | 142 | Death[12] | 0.271 | 0.86 | 0.239 | 0.032 | 31 | 5 |
| Statins+SGLT2i | 25 877 | Death (6252) | 0.359 | 0.80 | 0.299 | 0.059 | 17 | 1539 |
| RAASi+SGLT2i | 10 560 | Death (2363) | 0.327 | 0.82 | 0.277 | 0.050 | 20 | 532 |
| Statins+RAASi | 142 | CVD death[5] | 0.100 | 0.86 | 0.086 | 0.014 | 72 | 2 |
| Statins+SGLT2i | 25 877 | CVD death (4331) | 0.268 | 0.79 | 0.219 | 0.049 | 20 | 1273 |
| RAASi+SGLT2i | 10 560 | CVD death (1539) | 0.228 | 0.82 | 0.191 | 0.037 | 27 | 393 |
| Statins+RAASi | 146 | MACE[10] | 0.181 | 0.65 | 0.121 | 0.060 | 17 | 9 |
| Statins+SGLT2i | 26 738 | MACE (6728) | 0.368 | 0.65 | 0.257 | 0.111 | 9 | 2974 |
| RAASi+SGLT2i | 10 621 | MACE (2344) | 0.325 | 0.71 | 0.243 | 0.082 | 12 | 881 |
| Eligible for triple therapy | | | | | | | | |
| All therapies | 20 420 | Death (6661) | 0.464 | 0.75 | 0.374 | 0.090 | 11 | 1827 |
| All therapies | 20 420 | CVD death (3709) | 0.293 | 0.74 | 0.228 | 0.065 | 15 | 1339 |
| All therapies | 21 836 | MACE (5604) | 0.387 | 0.54 | 0.233 | 0.154 | 7 | 3349 |

The outcome death is death by any cause. Treatment with all therapies indicates simulated treatment with a combination of statins, RAASi, and SGLT2i. The probability of an event observed in the given subpopulation during follow-up. The estimated probability of event is the probability of an event when each given treatment is simulated using the HR for monotherapy reported in the relevant meta-analysis or the HR calculated under the assumption of a multiplicative relative effect of using treatments in combination. The absolute difference in risk is the difference in the observed probability of an event and the estimated probability of an event. The number needed to treat is the number of patients that need to be treated to prevent one event. CVD, cardiovascular disease; MACE, major adverse cardiovascular events; RAASi, renin-angiotensin aldosterone inhibitors; SGLT2i, sodium-glucose cotransporter-2 inhibitors.

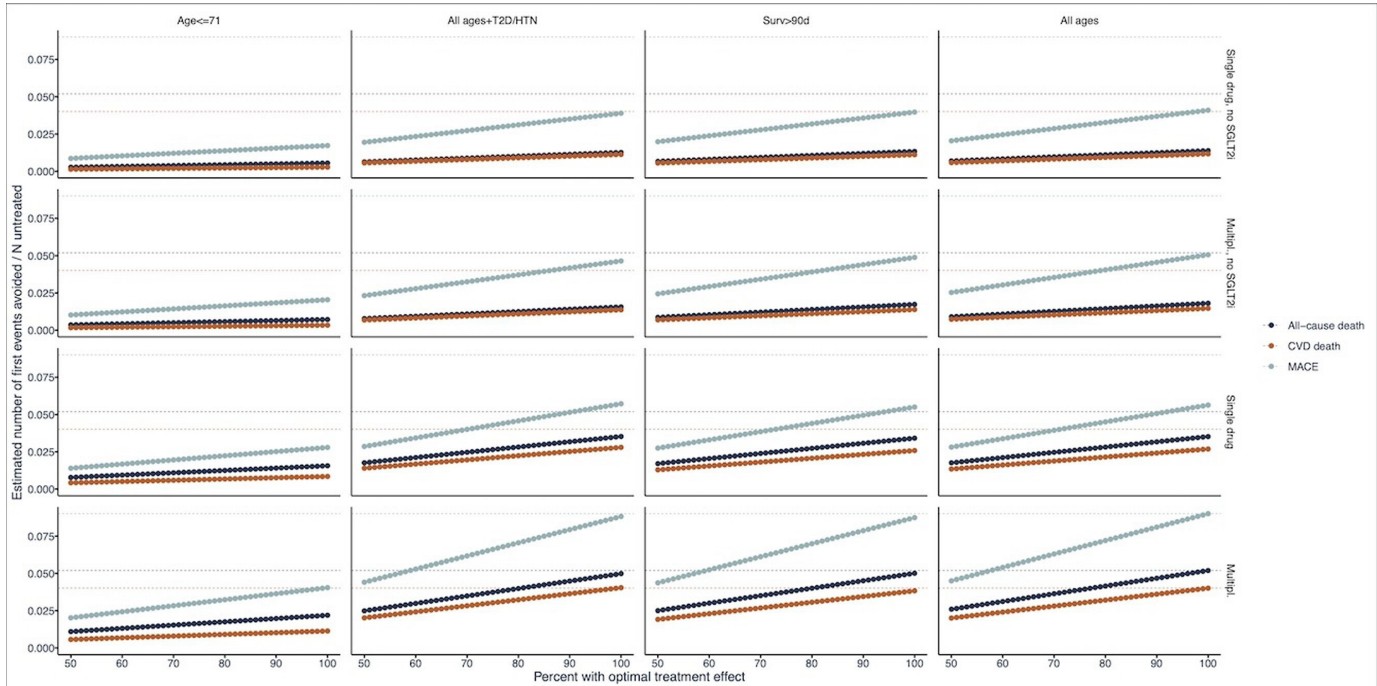

**Figure 4** Sensitivity analyses demonstrating variable treatment effects in less-than-optimal settings. Dashed, horizontal lines indicate optimal results (multiplicative effects of all drugs assumed, using all drugs and all age groups). Multipl., multiplicative effects of all drugs assumed; Multipl., no SGLT2i, multiplicative effects of all drugs assumed, but HR for SGLT2i set to 1; single drug, no multiplicative effects of drugs assumed, only the HR from the drug with the largest effect counted, HR for all other drugs set to 1; age ≤71, results only from a subset of persons in the lowest fourth of age (71 years); all ages+T2D/HTN, results only from a subset of persons with type-2 diabetes or hypertension; surv >90 days, results only from the subset that survived more than 90 days after index.

CKD treatment.[7] Lowering of blood pressure and lipids in patients with CKD has robust evidence for the prevention of both cardiovascular disease events and death,[22 23 30] and evidence is accumulating that SGLT2i reduces the risk of cardiorenal complications (heart failure and CKD) and all-cause mortality in all patients with CKD, irrespective of whether they have diabetes or not.[12 13 31] Combinations of these treatments are likely to have multiplicative protective effects.[25 26] Therefore, there is little reason, apart from side effects, to not use all three treatments in combination from the time of CKD diagnosis. On the contrary, RAASi and SGLT2i seem to have additive effects when used in combination. In the current study, adding SGLT2i to the treatment regimen provided the largest contribution to the combined treatment effect of the scenarios tested (figure 4).

While greater use of evidence-based treatment is likely the clearest opportunity for improving outcomes in CKD, deficiencies in the diagnosis of CKD also merit addressing. It is likely that the laboratory analyses used for CKD definitions in the present study are routinely performed for an array of conditions and complaints, but unless the caring physician is specifically looking for kidney disease, the historical data needed to make a CKD diagnosis may be overlooked. Although our findings are exclusively based on Swedish healthcare data, the low rate of CKD diagnosis in patients who fulfil CKD criteria has been described in numerous other healthcare systems

globally.[1] Future research might trial systems where eGFR and UACR values in databases are automatically monitored, notifying caring physicians if a patient presents abnormal measurements meeting the criteria for CKD. This concept has already been implemented in acute kidney injury, where electronic alert systems monitor electronic health records.[32 33] An automated system, such as that, may be the most reliable way to address the issue of human oversight, the likely reason so many cases of CKD went undiagnosed. Indeed, automated systems could theoretically be applied in multiple scenarios, from small databases within private health clinics to large national or regional databases, providing the capacity for continuous population-wide opportunistic screening; as compared with the current system, in which screening is only recommended in high-risk populations (ie, diabetes, hypertension and previously known cardiovascular disease).[34]

### Strengths and limitations

This study had access to health registries and electronic health records detailing most incidences of healthcare and medication use by residents in two of Sweden's largest regions, representing over a third of the national population. Despite its strength in its capacity to identify patients relevant to the study aims and to characterise the subsequent, large study population, several limitations must be acknowledged.

The identification of undiagnosed CKD was dependent on patients providing samples for laboratory analyses. Since only people with suspected or known other diseases have provided the samples necessary to make a laboratory-based assessment of CKD status, it is likely that this study underestimates the number of patients with undiagnosed CKD. It is possible that some diagnosing has taken place outside of the used data source. This would instead lead to an underestimation of the number of patients with a diagnosis of CKD. The follow-up period varied between patients, and in some cases, patients may have been enrolled towards the end of the inclusion period, leaving little opportunity for a diagnosis of CKD in healthcare. However, the methods used take person-time and censoring into account. It must also be acknowledged that this study could not assess if patients from a specific socioeconomic background or ethnicity were disproportionally affected by the deficiencies in CKD-related healthcare.

Simulations of guideline-directed pharmacological therapy assumed that treatment was initiated at the time CKD was identified using laboratory measurements and that adherence to treatment would be the same as that obtained in clinical trials. While indicating the full preventive potential, such efficiency and success in treatment are unlikely in routine clinical practice. The simulations assumed that the treatment effects observed in meta-analyses of each medication of interest were replicable in this study's untreated populations and that conjoint effects on outcomes did not deviate from multiplicativity, which has been demonstrated for RAASi and statins,[25 26] but not robustly for SGLT2i in relation to the other two drug classes.

Finally, the high age of the population merits some discussion. It can be debated if the age-related fall in eGFR should be viewed as glomerular senescence rather than kidney disease that uniformly requires treatment and if age-agnostic diagnostic thresholds should be used.[35] In the absence of guideline recommendations for treating elderly persons differently, we have taken a similar approach as one would to hypertension or heart failure and have assumed similar potential relative gains for all ages. Sensitivity analyses limiting the sample to younger persons provided less than half of the potential effect on the absolute scale than the analyses of the whole age distribution.

## CONCLUSION

CKD remained undiagnosed in two out of three patients over half a decade in routine healthcare, despite readily available data indicating its development. This highlights the need for automated systems that continuously monitor electronic health records and alert physicians when CKD has developed in a patient; eliminating any reliance on the physician to, first, decide to search for the disease and then make a diagnosis. Those who do receive a diagnosis get insufficient guideline-directed pharmacological therapy with statins, RAASi and SGLT2i, collectively potentiating a situation where CKD won't be addressed in healthcare until more advanced disease develops. Despite these lost opportunities, this study indicates that there is significant potential to substantially improve outcomes in CKD with stricter adherence to clinical practice guidelines by both the treating physicians and the patients.

**Author affiliations**
[1]Department of Medical Sciences, Clinical Epidemiology, Uppsala University, Uppsala, Sweden
[2]The George Institute for Global Health, University of New South Wales, Sydney, New South Wales, Australia
[3]Cardiology Unit, Department of Medicine, Karolinska Institutet, Solna, Stockholm, Sweden
[4]Capio S:t Görans Hospital, Stockholm, Sweden
[5]Department of Clinical Sciences, Danderyd Hospital, Karolinska Institute, Stockholm, Sweden
[6]Center for Diabetes, Academic Specialist Center, Region Stockholm, Stockholm, Sweden
[7]Cardiovascular, Renal and Metabolism, Medical Department, BioPharmaceuticals, AstraZeneca PLC, Oslo, Norway
[8]Sence Research AB, Uppsala, Sweden
[9]Department of Medical Sciences, Renal Medicine, Uppsala University, Uppsala, Sweden
[10]Uppsala Clinical Research Center, Uppsala, Sweden
[11]Division of Family Medicine, Department of Neurobiology, Care Sciences and Society, Karolinska Institutet, Stockholm, Sweden
[12]School of Health and Social Studies, Dalarna University, Falun, Dalarna, Sweden

**Acknowledgements** The authors thank Jordan Loader PhD, of Sence, Uppsala, Sweden, for providing medical writing support, which was funded by AstraZeneca, Stockholm, Sweden, in accordance with Good Publication Practice (GPP3) guidelines (http://www.ismpp.org/gpp3). The authors also thank Susanna Jerström and Pål Hasvold from AstraZeneca for valuable comments on the manuscript.

**Contributors** JS, AN, SK, JB, SG, TC, MKS and JÄ developed the concept of the study. SG and TC performed all data curation and statistical analyses. JS, AN, SK, SG, TC, MKS and JÄ participated in data interpretation. JS drafted the original manuscript, with critical intellectual input from AN, SK, JB, SG, TC, MKS and JÄ. JS is theguarantor and accepts full responsibility for the work, had access to the data, and controlled the decision to publish.

**Funding** The study was fully sponsored by AstraZeneca N/A.

**Competing interests** JS reports stock ownership in Anagram kommunikation AB and Symptoms Europe AB. AN has participated as a consultant on advisory board committees and educational activities with Astra Zeneca, Novo Nordisk, Boehringer Ingelhim and Eli Lilly. SK has participated as a consultant on advisory board committees and educational activities with AstraZeneca and Novo Nordisk. JB is an employee of AstraZeneca. TC is an employee and stockowner in Sence Research AB (an independent company in epidemiology and outcomes research) that received funding from AstraZeneca for statistical analysis within this research project. MKS has participated as a consultant on advisory board committees and educational activities with Amgen, AstraZeneca, Boehringer Ingelheim, GSK and NovoNordisk. JÄ has received payment/honoraria from AstraZeneca and Novartis; participated on a data safety monitoring/advisory board for AstraZeneca, Astella and Boehringer Ingelheim. The other authors report no competing interests.

**Patient and public involvement** Patients and/or the public were not involved in the design, or conduct, or reporting, or dissemination plans of this research.

**Patient consent for publication** Not applicable.

**Ethics approval** This study involves human participants and was approved by the Swedish Ethical Review Authority (approval numbers 2020-03850 and 2020-06716). Not required in Sweden for use of registry data.

**Provenance and peer review** Not commissioned; externally peer reviewed.

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
