## [Reviewer comments · BMJ Open]

ARTICLE DETAILS

TITLE (PROVISIONAL)	Are there lost opportunities in chronic kidney disease? A region-wide cohort study
AUTHORS	Sundström, Johan; Norhammar, Anna; Karayiannides, Stelios; Bodegård, Johan; Gustafsson, Stefan; Cars, Thomas; Eriksson Svensson, Maria; Ärnlöv, Johan

VERSION 1 – REVIEW

REVIEWER	Brimble, Kenneth Scott McMaster University
REVIEW RETURNED	12-May-2023

GENERAL COMMENTS	The manuscript by Sundstrom and colleagues addresses the question of what the potential impact of treating patients with newly diagnosed CKD would be if statins, RAASi and SGLT2i were all used. They suggest a remarkably low NNT over 5 years of 8 to prevent a death in that time period. They also note that the diagnosis of CKD is infrequently made and when made, significantly delayed. I believe this to be important work that would have the potential to influence practice, particularly in primary care. Overall, it is well written. I have a number of questions/concerns that I would like addressed before I would consider the manuscript ready for publication. - It was unclear to me if authors had access to all primary care health records where initial CKD diagnosis is likely to be made.- does primary care use this particular eGFR equation? I am not familiar with it and wonder if the primary care providers are and also if the eGFR is auto-reported (and if so using what equation).- I would like to see sensitivity analyses with reasonable estimates of adherence/tolerability to the agents. In fact, it would be more reasonable to do the main analysis using an evidence-informed estimate of adherence and tolerability and then provide some sensitivity analyses providing estimates around that initial value.- FYI the references in Table 2 as numbered do not match the actual reference list numbering system.- I did not understand from the text how the references were chosen for determining HRs for the various agents and then from these how a final HR was chosen for a given agent (an average?). For example, there are 2 references listed for RAASi monotherapy. The first one by Qin was a meta-analysis of observational studies when there are meta-analyses of RAASi trials in CKD that included only RCTs which would be more appropriate (acknowledging most trials were vs placebo not another active agent). I did not look at the second one specifically. I am also concerned that the efficacy of these agents is overstated
---

	as the current population was primarily non-proteinuric and I suspect of lower CV risk than those in clinical trials. I am not sure how much that is an issue here. - the findings are mostly driven by the estimated effect of using SGLT2i which were little used during this time period. I won't quibble too much about the estimate that about 38% of patients would have been eligible for its use (sounds high to me) but I do think it could be more strongly stated that the evidence for their use (and presumably availability) in many of these patients would have been lacking or only had recently emerged. You state: "Given that SGLT2i have only recently been included in clinical practice guidelines after the study period,(7,11,12) low use of these agents was in line with expectations. Hence, our data clearly show that opportunities are lost in both the diagnosis and treatment of CKD." The final sentence doesn't really follow from the previous one. - there is an "M" missing from "monotherapy" in the first column of Table 2. Thanks.
--	--

REVIEWER	Norris, KC University of California, Los Angeles
REVIEW RETURNED	04-Aug-2023

GENERAL COMMENTS	Thank you for the opportunity to review this perspective entitled "Lost opportunities in chronic kidney disease". This is a well written manuscript that sought to identify the opportunities for diagnosis of chronic kidney disease (CKD) and the prevention of its adverse outcomes by examining data of residents in the Stockholm and Skåne regions of Sweden between January 1st, 2015, and December 31st, 2020. The authors found only 1/3 of nearly 100,000 residents with undiagnosed CKD during the study period received a subsequent diagnosis of CKD after five years. In simulations they estimated 25% of deaths and 38% of major adverse cardiovascular events could have been avoided if every patient that did not use an indicated medication for their laboratory-confirmed CKD was treated with guideline-directed pharmacological therapy for CKD. The authors then conclude they identified a critical under-diagnosis and under-treatment of CKD in this large contemporary population, with substantial potential to improve CKD outcomes and reduce its burden by treating patients early with guideline-directed pharmacological therapy. Comments This is addressing an important question of undiagnosed CKD and clinical implications of the use or non-use of CKD guideline-directed pharmacological therapy. The methodological approaches and assessment of key medications appear reasonable. The major concern is the age of the population (mean age 77.7 yrs) and the lack of consideration of age-related fall in eGFR due to glomerular senescence rather than kidney disease.
---

	With increasing age providers may be making a determination of age-related fall in the eGFR and not intrinsic kidney disease. Therefore, a decision to not make a diagnosis of chronic kidney disease and not initiate therapy for such may in fact be well grounded. As such one approach could be to conduct the analysis with censoring patients > 65 years of age who have a reduced eGFR but no proteinuria/albuminuria – maybe for those without data on urine protein/albumin assess for major CKD risk factors as co-morbidities that might be suggestive of renal disease (e.g. treated DM or HTN). This would give a better insight into the actual magnitude undiagnosed CKD by eliminating patients/residents with potential age-related reduction in eGFR. There might be other approaches as well such as the use of an age adapted definition of CKD for elderly people of <45 ml/min per 1.73 m², a level associated with increased mortality risk in elderly. Delanaye P, Jager KJ, Bökenkamp A, Christensson A, Dubourg L, Eriksen BO, Gaillard F, Gambaro G, van der Giet M, Glassock RJ, Indridason OS, van Londen M, Mariat C, Melsom T, Moranne O, Nordin G, Palsson R, Pottel H, Rule AD, Schaeffner E, Taal MW, White C, Grubb A, van den Brand JA. CKD: A Call for an Age-Adapted Definition. J Am Soc Nephrol. 2019 Oct;30(10):1785-1805. Glassock RJ, Delanaye P, Rule AD. Should the definition of CKD be changed to include age-adapted GFR criteria? YES. Kidney Int. 2020 Jan;97(1):34-37. Levey AS, Inker LA, Coresh J. "Should the definition of CKD be changed to include age-adapted GFR criteria?": Con: the evaluation and management of CKD, not the definition, should be age-adapted. Kidney Int. 2020 Jan;97(1):37-40.
--	---

REVIEWER	Rutter, Charlotte LSHTM, Medical Statistics
REVIEW RETURNED	22-Jan-2024

GENERAL COMMENTS	Overall: This seems to be a very valuable study, identifying potential opportunities to improve outcomes through the use of electronic health records to flag potential CKDu, and targeting improved adherence to treatment regimens. However, there are some issues in the analysis that may overstate the benefits, and some methods which need clarification. Introduction: It would be good to differentiate more between the problem of under-diagnosis vs under-treatment in the current literature e.g., when you say “low proportions of patients with any degree of CKD are treated...” is this doctor diagnosed or based on laboratory findings? Methods:
--

Please expand on the MSM models used, what were the parameterisations of the transitions, did you assume proportional hazards etc. This will help researchers reproduce comparable analyses in other populations. It is ok if this is in a supplement.

In the statistical analysis section it is stated that CKD was identified using primarily eGFR measures if available and UACR if eGFR was missing. This does not match up to the definition in the study population section which states 2 abnormal eGFR and/or 2 abnormal UACR measurements were used to define CKDu. Please clarify.

For the simulations, I don't think the assumption that the reported treatment effect was replicable here has been sufficiently justified e.g., have you accounted for different age distributions between the meta analyses and your study, particularly when looking at all-cause mortality? Have you taken into account that your sample was from people already undergoing hospital tests for other conditions? I have similar reservations for the use of multiplicative effects, particularly for three treatments.

Results:

It is not clear where your "HR from meta-analysis" column in Table 2 came from. There is a list of different meta-analyses in Table S1 but how did you choose which to use? Please also provide the CIs around your HRs and use them to add CIs around the estimated number of events prevented.

I would like to see some sensitivity analysis around different levels of treatment effect (including the effect of multiple treatments), adherence to treatment, and a time lag between identifying CKDu and the patient receiving the treatment. I know these things were discussed in the limitations but the present quoted figures are too "perfect" to be realistic but there is nothing to compare them to.

The majority of your improvement seems to come from adding the newer drug SGLT2i (as you state, very few people were using it). It would be good to see the saving split out separately by 1. the gap in the established treatment protocol (of RAASI and Statins) and 2. the additional roll out of SGLT2i. This would show where the focus should be.

Discussion:

Please include some thoughts on why the patients identified using UACR are so different from those identified using eGFR i.e., they are younger, with diabetes, higher BMI, more males. This stands out in Table 1.

It really would be good to try to split out the potential improvements from earlier diagnosis, versus that from adherence to treatment guidelines. Sometimes the discussion jumps between the two.

Conclusion:

The conclusion highlights important gaps in renal care, even in a high-income European country with a comprehensive public healthcare system, and suggests a potential, achievable method for improvement. This could also be useful for policymakers in other settings with automated record keeping.

VERSION 1 – AUTHOR RESPONSE

Comments from Reviewer 1 (Dr. Kenneth Scott Brimble, McMaster University)

Comment #1

It was unclear to me if authors had access to all primary care health records where initial CKD diagnosis is likely to be made.

Response to Comment #1

The study sample included patients from two large regions in Sweden, Stockholm and Skåne. For Region Stockholm, data for ICD10 diagnoses (including diagnoses for chronic kidney disease) are extracted from both public and private primary caregivers with high coverage. For Region Skåne, we have high coverage for primary care diagnoses from public primary caregivers, but limited coverage for diagnoses from private primary caregivers. We estimate the proportion of public primary healthcare givers to be 60-70% in Region Skåne.

Comment #2

Does primary care use this particular eGFR equation? I am not familiar with it and wonder if the primary care providers are and also if the eGFR is auto-reported (and if so using what equation).

Response to Comment #2

The revised Lund-Malmö equation is validated, recommended, and gaining in popularity. The CKD-EPI equation (and some others) has been criticized for including a variable for self-reported race, which is prone to error and undertreatment of people of colour. Some recent attempts to remove race from the equations can be found in NEJM 2021;385:1804, NEJM 2021;385:1750, and NEJM 2021;385:1737. In Sweden, the Lund-Malmö equation is nationally recommended (https://www.equalis.se/media/gw2jimdr/s016_beräkning-av-egfr-från-kreatinin-och-cystatin-c_1-2.pdf) and the equation built into EHR systems and provided automatically for all public healthcare providers. Hence is particularly relevant to use the Lund-Malmö equation for our study as clinicians in primary care base their assessment of GFR on this equation. It would make less sense to assess opportunities lost if we had calculated GFR differently than was done in the clinical situation.

Comment #3

I would like to see sensitivity analyses with reasonable estimates of adherence/tolerability to the agents. In fact, it would be more reasonable to do the main analysis using an evidence-informed estimate of adherence and tolerability and then provide some sensitivity analyses providing estimates around that initial value.

Response to Comment #3

The effect estimates from the meta-analyses of randomized controlled trials are based on intention-to-treat (not per-protocol) analyses, hence trial effects are not based on perfect adherence either. However, we agree that the adherence in a real-world situation might be even poorer than in a trial

and a lower treatment effect is possible. To address this, as part of the simulation, we conducted multiple sensitivity analyses allowing consideration of a treatment effect between 50-100% of the treatment effect obtained in clinical trials (new **Figure 4**, also below, please also see response to Comment #4 by Reviewer #3 below for a full description of these sensitivity analyses).

Comment #4

FYI the references in Table 2 as numbered do not match the actual reference list numbering system.

Response to Comment #4

Table 2 in the manuscript didn't/doesn't contain any references.

Comment #5

I did not understand from the text how the references were chosen for determining HRs for the various agents and then from these how a final HR was chosen for a given agent (an average?). For example, there are 2 references listed for RAASi monotherapy. The first one by Qin was a meta-analysis of observational studies when there are meta-analyses of RAASi trials in CKD that included only RCTs which would be more appropriate (acknowledging most trials were vs placebo not another active agent). I did not look at the second one specifically. I am also concerned that the efficacy of these agents is overstated as the current population was primarily non-proteinuric and I suspect of lower CV risk than those in clinical trials. I am not sure how much that is an issue here.

Response to Comment #5

As part of this review process, we have updated our analysis and have selected one broad systematic review and meta-analysis for each drug. Importantly, we have used a more recent meta-analysis for SGLT2i treatment, which includes new trial data (e.g., EMPA-Kidney) that had become available and analysed in meta-analyses since the initial submission of our study. We now have only used meta-analyses of randomized controlled trials. The three meta-analyses, one for each treatment of interest, were selected as each provided a comprehensive compilation of trials and assessed the

effect of either statin, RAASi, or SGLT2i treatment on the risk of all cause death, cardiovascular death, and MACE, the three outcomes of interest. We have edited the text in the statistical analysis section to provide this rationale:

"Hazard ratios from several large systematic reviews and meta-analyses,(1–3) which provided a comprehensive compilation of randomized controlled trials that evaluated the effect of statin, RAASi, or SGLT2i treatment on all-cause death, cardiovascular death, and MACE in patients with CKD, were used to calculate the probability of five-year survival for those outcomes, simulating the use of guideline-directed pharmacological therapy in the total number of patients untreated with that specific therapy." (Page 7)

For simulations of monotherapy, the relevant hazard ratios were extracted directly from each meta-analysis. For simulations of dual therapy and triple therapy, hazard ratios from were calculated using the hazard ratios from the three meta-analyses under the assumption that there was no deviation from a multiplicative relative effect of using treatments in combination. We edited the text to make this clearer:

"It was assumed that the treatment effect reported by each meta-analysis was replicable in each simulation and that medication adherence was similar to that obtained in clinical trials from the date that CKD was identified using laboratory measurements. For simulations of monotherapy, the relevant hazard ratio extracted from each meta-analysis was used. For combination therapy, hazard ratios were calculated using the hazard ratios extracted from the three meta-analyses under the assumption that there was no deviation from a multiplicative relative effect of using treatments in combination (i.e., no interaction between medications), as shown previously.(4–6)"

Comment #6

The findings are mostly driven by the estimated effect of using SGLT2i which were little used during this time period. I won't quibble too much about the estimate that about 38% of patients would have been eligible for its use (sounds high to me) but I do think it could be more strongly stated that the evidence for their use (and presumably availability) in many of these patients would have been lacking or only had recently emerged. You state: "Given that SGLT2i have only recently been included in clinical practice guidelines after the study period,(7,11,12) low use of these agents was in line with expectations. Hence, our data clearly show that opportunities are lost in both the diagnosis and treatment of CKD." The final sentence doesn't really follow from the previous one.

Response to Comment #6

That's fair; we have now changed the "Hence" to "In sum", as that sentence was intended to summarize the whole section above it, not the sentence immediately preceding it. We have also added sensitivity analyses specifically investigating the contribution of SGLT2i to the treatment effect, and added wording about them to Results:

"The sensitivity analysis showed that the potential to prevent all-cause deaths, cardiovascular deaths, and MACE decreased as the proportion that received optimal treatment decreased (Figure 4). The potential to prevent these events was most notably lower in younger persons and when treatment with SGLT2i was not included as part of combination therapy."

And Discussion:

“In the current study, adding an SGLT2i to the treatment regimen provided the largest contribution to the combined treatment effect of the scenarios tested (Figure 4).”

Comment #7

There is an "M" missing from "monotherapy" in the first column of Table 2.

Response to Comment #7

Thank you for highlighting this error. We have corrected it.

Comments from Reviewer 2 (Prof. KC Norris, University of California, Los Angeles)

Comment #1

The major concern is the age of the population (mean age 77.7 yrs) and the lack of consideration of age-related fall in eGFR due to glomerular senescence rather than kidney disease.

With increasing age providers may be making a determination of age-related fall in the eGFR and not intrinsic kidney disease. Therefore, a decision to not make a diagnosis of chronic kidney disease and not initiate therapy for such may in fact be well grounded.

As such one approach could be to conduct the analysis with censoring patients > 65 years of age who have a reduced eGFR but no proteinuria/albuminuria – maybe for those without data on urine protein/albumin assess for major CKD risk factors as co-morbidities that might be suggestive of renal disease (e.g. treated DM or HTN).

This would give a better insight into the actual magnitude undiagnosed CKD by eliminating patients/residents with potential age-related reduction in eGFR.

There might be other approaches as well such as the use of an age adapted definition of CKD for elderly people of <45 ml/min per 1.73 m², a level associated with increased mortality risk in elderly.

Delanaye P, Jager KJ, Bökenkamp A, Christensson A, Dubourg L, Eriksen BO, Gaillard F, Gambaro G, van der Giet M, Glassock RJ, Indridason OS, van Londen M, Mariat C, Melsom T, Moranne O, Nordin G, Palsson R, Pottel H, Rule AD, Schaeffner E, Taal MW, White C, Grubb A, van den Brand JAIG. CKD: A Call for an Age-Adapted Definition. *J Am Soc Nephrol.* 2019 Oct;30(10):1785-1805.

Glassock RJ, Delanaye P, Rule AD. Should the definition of CKD be changed to include age-adapted GFR criteria? YES. *Kidney Int.* 2020 Jan;97(1):34-37.

Levey AS, Inker LA, Coresh J. "Should the definition of CKD be changed to include age-adapted GFR criteria?": Con: the evaluation and management of CKD, not the definition, should be age-adapted. *Kidney Int.* 2020 Jan;97(1):37-40.

Response to Comment #1

This is an important topic, and there is no simple answer. The pros and cons of age-specific CKD definitions have been widely debated in recent decades. Despite strong arguments favoring a revision of the CKD definition in this regard this debate has yet to have a substantial impact in clinical guidelines or the definition of CKD in clinical practice (the recent KDIGO definition is age-agnostic, for example: <https://kdigo.org/guidelines/ckd-evaluation-and-management/>).

For perspective, viewing higher systolic blood pressure as a part of normal aging and having laxer treatment targets for older persons is a thing of the past. And, apart from heart transplantation, we would not consider treating 77-year-old heart failure patients less aggressively than younger heart failure patients simply because of their age (the mean age of many heart failure populations is the same as in this CKD population).

That said, the topic merits discussion, and we have included a paragraph to that effect, also including the new sensitivity analyses:

“Finally, the high age of the population merits some discussion. It can be debated if the age-related fall in eGFR should be viewed as glomerular senescence rather than kidney disease that uniformly requires treatment, and if age-agnostic diagnostic thresholds should be used .(7) In the absence of guideline recommendations for treating elderly persons differently, we have taken a similar approach as one would to hypertension or heart failure, and have assumed similar potential relative gains for all ages. Sensitivity analyses limiting the sample to younger persons provided less than half of the potential effect on the absolute scale than the analyses of the whole age distribution.”

Comments from Reviewer 3 (Dr. Charlotte Rutter, LSHTM)

Response to Comment #1

It would be good to differentiate more between the problem of under-diagnosis vs under-treatment in the current literature e.g., when you say “low proportions of patients with any degree of CKD are treated...” is this doctor diagnosed or based on laboratory findings?

Response to Comment #1

Those studies used laboratory-confirmed CKD for most analyses. We have now clarified that:

“Data from recent large cohort studies suggest that low proportions of patients with any degree of laboratory-confirmed CKD are diagnosed with CKD or treated with these drugs”.

Comment #2

Please expand on the MSM models used, what were the parameterisations of the transitions, did you assume proportional hazards etc. This will help researchers reproduce comparable analyses in other populations. It is ok if this is in a supplement.

Response to Comment #2

The Aalen-Johansen estimator of transition probabilities was used in a multi-state model adjusting for age (categorical [18,45) [45,65) [65,80) [80,Inf)) and sex. Death and MACE were set as absorbing states and transient states were "Undiagnosed+Untreated", "Undiagnosed+Treated", "Diagnosed+Untreated", "Diagnosed+Treated". Once diagnosed, transitioning back to an undiagnosed state was obviously not possible. The Aalen-Johansen estimator does not require assumptions of proportionality. We added the following to Methods:

"The Aalen-Johansen estimator of transition probabilities was used in a multi-state model adjusting for age (categorical [18,45) [45,65) [65,80) [80,Inf)) and sex."

Comment #3

In the statistical analysis section it is stated that CKD was identified using primarily eGFR measures if available and UACR if eGFR was missing. This does not match up to the definition in the study population section which states 2 abnormal eGFR and/or 2 abnormal UACR measurements were used to define CKD. Please clarify.

Response to Comment #3

As described in the 'study population' section, eGFR and/or UACR measurements were used to identify patients with CKD. However, in figures for multi-state models and all tables where patients were stratified according to the severity of their CKD, stratification was completed primarily using eGFR measurements. If eGFR data were not available, we used UACR data for stratification purposes. We recognize how the original wording in the statistical analysis section would have made the methodology for identifying CKD confusing. We have therefore edited the sentence in the statistical analysis section to clarify this:

"In figures for the multi-state models and all tables where patients are stratified by the severity of their CKD, patients were stratified primarily using eGFR measurements. If eGFR data were normal or unavailable, UACR measurements were used to stratify the patients." (Page 6)

Comment #4

For the simulations, I don't think the assumption that the reported treatment effect was replicable here has been sufficiently justified e.g., have you accounted for different age distributions between the meta-analyses and your study, particularly when looking at all-cause mortality? Have you taken into account that your sample was from people already undergoing hospital tests for other conditions? I have similar reservations for the use of multiplicative effects, particularly for three treatments.

Response to Comment #4

We have now included several sensitivity analyses to account for variable treatment effects due to differences in the population (e.g., age, comorbidities), the influence of treatment with SGLT2i, or less than perfect medication adherence (new **Figure 4**, below).

Dashed, horizontal lines indicate optimal results (multiplicative effects of all drugs assumed, using all drugs and all age groups). *Multipl.*, multiplicative effects of all drugs assumed; *Multipl., no SGLT2i*, multiplicative effects of all drugs assumed, but HR for SGLT2i set to 1; *Single drug*, no multiplicative effects of drugs assumed, only the HR from the drug with the largest effect counted, HR for all other drugs set to 1; *Age<=71*, results only from subset of persons in lowest fourth of age (71 years); *All ages+T2D/HTN*, results only from subset of persons with type-2 diabetes or hypertension, which would already have other indications for some of the drugs and hence be more likely to get treatment (less controversial to consider for combination preventive treatment); *Surv>90d*, results only from subset that survived more than 90 days after index, which would be more likely to be considered for preventive treatment. The X axis illustrates percent of the untreated that get optimal treatment effect (the intention-to-treat effect obtained in clinical trials); i.e. if there are delays in initiating treatment, and/or adherence is less than that obtained in clinical trials, and/or other reasons for not expecting optimal treatment effect, one can consider a number less than 100.

These sensitivity analyses have now been described in Methods:

“Multiple sensitivity analyses were conducted to account for variable treatment effects due to differences in the population, including a subgroup of younger age (with higher likelihood to be considered for preventive treatment); a subgroup with type-2 diabetes or hypertension (which would already have other indications for some of the drugs and hence be more likely to be considered for preventive treatment); and a subgroup that survived more than 90 days after index (which would be more likely to be considered for preventive treatment than very sick patients); and variable treatment effects due to differences in the treatment, including combinations of assumptions of multiplicative effects of all drugs assumed, but HR for SGLT2i set to 1; and assumptions of no multiplicative effects, only the HR from the drug with the largest effect counted, HR for all other drugs set to 1. These analyses also investigated the influence of less-than-optimal treatment effect (the intention-to-treat effect obtained in clinical trials); i.e. if there are delays in initiating treatment, and/or adherence is less than that obtained in clinical trials, and/or other reasons for not expecting optimal treatment effect, a number less than 100 could be considered.”

Results:

“The sensitivity analysis showed that the potential to prevent all-cause deaths, cardiovascular deaths, and MACE decreased as the proportion that received optimal treatment decreased (Figure 4). The potential to prevent these events was most notably lower in younger persons and when treatment with SGLT2i was not included as part of combination therapy.”

And Discussion:

“Sensitivity analyses limiting the sample to younger persons provided less than half of the potential effect on the absolute scale than the analyses of the whole age distribution.” and “In the current study, adding an SGLT2i to the treatment regimen provided the largest contribution to the combined treatment effect of the scenarios tested (Figure 4).”

Comment #5

It is not clear where your “HR from meta-analysis” column in Table 2 came from. There is a list of different meta-analyses in Table S1 but how did you choose which to use? Please also provide the CIs around your HRs and use them to add CIs around the estimated number of events prevented.

Response to Comment #5

Table S1 includes a summary of effects estimates from relevant meta-analyses of observational studies and/or randomized controlled trials, which was constructed to support the introduction. We selected three meta-analyses, one for each treatment of interest, which provided a comprehensive compilation of trials assessing the effect of either statin, RAASi, or SGLT2i treatment on the risk of all cause death, cardiovascular death, and MACE, the three outcomes of interest. These meta-analyses only assessed randomized controlled trials. The “HR from meta-analysis” column contains relevant hazard ratios that were extracted directly from each meta-analysis for simulations of monotherapy. For simulations of dual therapy and triple therapy, hazard ratios from were calculated using the hazard ratios extracted from the three meta-analyses under the assumption that there was no deviation from a multiplicative relative effect of using treatments in combination. We have edited the statistical analysis section to make it clearer where the “HR from meta-analysis” column comes from (**Page 7**).

We have provided the confidence intervals in Table 2 for the hazard ratios extracted from the meta-analyses, i.e., the hazard ratios used for simulations of monotherapy. Fair confidence intervals couldn't be calculated for the hazard ratios that were used for simulations of combination therapy. Similarly, confidence intervals couldn't be calculated for the number of events prevented.

Comment #6

I would like to see some sensitivity analysis around different levels of treatment effect (including the effect of multiple treatments), adherence to treatment, and a time lag between identifying CKD and the patient receiving the treatment. I know these things were discussed in the limitations but the present quoted figures are too “perfect” to be realistic but there is nothing to compare them to.

Response to Comment #6

Please see response to comment #4; we have now included multiple sensitivity analyses (new **Figure 4**, below). This serves the purpose of addressing many of these factors simultaneously; i.e. a departure from “optimal treatment effect” could be due to any of or combinations of these factors.

Comment #7

The majority of your improvement seems to come from adding the newer drug SGLT2i (as you state, very few people were using it). It would be good to see the saving split out separately by 1. the gap in the established treatment protocol (of RAASI and Statins) and 2. the additional roll out of SGLT2i. This would show where the focus should be.

Response to Comment #7

The now included sensitivity analyses (above) demonstrates the reduction in the potential to prevent events when the treatment effect of SGLT2i has been set to a hazard ratio of 1 (i.e., no effect) for the rows labelled “no SGLT2i”. It can be noted that most of the potential comes from adding an SGLT2i. This has now been added to the Results and Discussion.

“In the current study, adding an SGLT2i to the treatment regimen provided the largest contribution to the combined treatment effect of the scenarios tested (Figure 4).”

Comment #8

Please include some thoughts on why the patients identified using UACR are so different from those identified using eGFR i.e., they are younger, with diabetes, higher BMI, more males. This stands out in Table 1.

Response to Comment #8

These differences in characteristics have been observed previously.(8) It may be speculated that the A2+A3 categories represent earlier stages of CKD. Interestingly, the rates over time of receiving a

CKD diagnosis was similar in stages S3a and A2, and stages S3b and A3, respectively. These differences in characteristics and these rates have now been added to the Results:

“At baseline, those with CKD stages A2 and A3 were younger, had higher BMI, were more often male, more often had diabetes, and more often had preventive drug treatment, than those with stages S3a to S5 (Table 1).” and “Notably, the rates over time of receiving a CKD diagnosis was similar in stages S3a and A2, and stages S3b and A3, respectively (Figure 1 and Figure 2).”

And to the Discussion:

“Patients with CKD stages A2 and A3 more often used SGLT2i than those with stages S3a to S5, both at baseline and during follow-up (Table 1 and Figure 3), explained by their more than twice as high prevalence of diabetes (Table 1).”

Comment #9

It really would be good to try to split out the potential improvements from earlier diagnosis, versus that from adherence to treatment guidelines. Sometimes the discussion jumps between the two.

Response to Comment #9

This is a bit tricky, as receiving a CKD diagnosis did not result in immediate initiation of guideline-directed pharmacological therapy in this study. This is commented twice in the first two paragraphs of the Discussion. We have now restructured the Discussion a little to keep the parts about earlier diagnosis together, and we have introduced that section thus:

“While greater use of evidence-based treatment is likely the clearest opportunity for improving outcomes in CKD, deficiencies in the diagnosis of CKD also merit addressing.”

Comment #10

The conclusion highlights important gaps in renal care, even in a high-income European country with a comprehensive public healthcare system, and suggests a potential, achievable method for improvement. This could also be useful for policymakers in other settings with automated record keeping.

Response to Comment #10

We thank the Reviewer for the thorough review and for these kind comments.

References

1. Hou W, Lv J, Perkovic V, Yang L, Zhao N, Jardine MJ, et al. Effect of statin therapy on cardiovascular and renal outcomes in patients with chronic kidney disease: a systematic review and meta-analysis. *Eur Heart J*. 2013 Jun;34(24):1807–17.
2. Balamuthusamy S, Srinivasan L, Verma M, Adigopula S, Jalandhara N, Jalandara N, et al. Renin angiotensin system blockade and cardiovascular outcomes in patients with chronic kidney disease and proteinuria: a meta-analysis. *Am Heart J*. 2008 May;155(5):791–805.
3. Mavrakanas TA, Tsoukas MA, Brophy JM, Sharma A, Gariani K. SGLT-2 inhibitors improve cardiovascular and renal outcomes in patients with CKD: a systematic review and meta-analysis. *Sci Rep*. 2023 Sep 23;13(1):15922.
4. Sundström J, Gulliksson G, Wirén M. Synergistic effects of blood pressure-lowering drugs and statins: systematic review and meta-analysis. *BMJ Evid Based Med*. 2018 Apr;23(2):64–9.
5. Kanukula R, Esam H, Sundström J, Rodgers A, Salam A. Does Co-administration of Antihypertensive Drugs and Statins Alter Their Efficacy and Safety? A Systematic Review and Meta-analysis of Randomized Controlled Trials. *J Cardiovasc Pharmacol*. 2019 Jun;73(6):352–8.
6. Wald DS, Law M, Morris JK, Bestwick JP, Wald NJ. Combination therapy versus monotherapy in reducing blood pressure: meta-analysis on 11,000 participants from 42 trials. *Am J Med*. 2009 Mar 1;122(3):290–300.
7. Delanaye P, Jager KJ, Bökenkamp A, Christensson A, Dubourg L, Eriksen BO, et al. CKD: A Call for an Age-Adapted Definition. *J Am Soc Nephrol*. 2019 Oct;30(10):1785–805.
8. Ji E, Kim YS. Prevalence of chronic kidney disease defined by using CKD-EPI equation and albumin-to-creatinine ratio in the Korean adult population. *Korean J Intern Med*. 2016 Nov;31(6):1120–30.

VERSION 2 – REVIEW

REVIEWER	Norris, KC University of California, Los Angeles
REVIEW RETURNED	27-Mar-2024
GENERAL COMMENTS	Thank you for addressing all comments
REVIEWER	Rutter, Charlotte LSHTM, Medical Statistics
REVIEW RETURNED	03-Apr-2024
GENERAL COMMENTS	I am happy with the revisions, and the responses covered all the points I raised. I particularly like the addition of the wide-ranging sensitivity analyses.